## RESEARCH ARTICLE

# Left behind in primary healthcare: A qualitative exploration of healthcare experiences of people with disabilities in Ethiopia

Desta Debalkie Atnafu [1,2]*, Hannah Kuper[1], Femke Bannink Mbazzi [1,3]

1 International Centre for Evidence in Disability, Department of Population Health, Faculty of Epidemiology and Population Health, London School of Hygiene and Tropical Medicine, London, United Kingdom, 2 Department of Health System Management and Health Economics, School of Public Health, Bahir Dar University, Bahir Dar, Ethiopia, 3 Disability Research Group, MRC/UVRI & LSHTM Uganda Research Unit, Entebbe, Uganda

* desta.debalkie@lshtm.ac.uk, destad2a@gmail.com

## Abstract

People with disabilities, who make up 1.3 billion globally, frequently face systemic exclusion from healthcare due to a range of barriers. This qualitative study explored the healthcare access experiences of 30 adults with disabilities in Bahir Dar City, Ethiopia, aiming to identify barriers, facilitators, and context-driven solutions. Both purposive and snowball sampling was conducted to identify participants. In-depth interviews were conducted in the local language. Data were analysed using reflexive thematic analysis in NVivo 14, guided by the Missing Billion Health System Framework service delivery components. The study identified five key themes each for barriers, facilitators, and coping strategies. Major barriers included low health literacy, unaffordable care, negative provider attitudes, inaccessible infrastructure, and lack of assistive technologies and rehabilitation services. Facilitators included family support, community-based health insurance, disability-sensitive training of healthcare workers, presence of a rehabilitation centre, and initiation of renovation infrastructure in model facilities. Participants proposed actionable strategies such as increasing awareness, insurance coverage, local production of assistive technologies, assigning personal assistants in health facilities, improving accessibility, and establishing disability units within governance structures. People with disabilities experienced persistent, intersecting barriers to healthcare access in Ethiopia. However, scaling disability-inclusive training, infrastructure improvemnts, and governance reforms—rooted in lived experience and aligned with human rights of people with disabilities—can help drive progress toward Universal Health Coverage.

**Data availability statement:** The data supporting the findings of this study are not publicly available due to ethical restrictions. However, data can be accessed upon reasonable request from London School of Hygiene & Tropical Medicine / Amhara Public Health Institute, at "researchdatamanagement@lshtm.ac.uk / admin@aphi.gov.et" respectively. This institutional body is independent of the study authors and will review data access requests in accordance with ethical and institutional guidelines.

**Funding:** This study was supported by the Younger Family Foundation (YFF) PhD Scholarship/Studentship. Desta Debalkie Atnafu is funded through an award from YFF, and Hannah Kuper is supported by a grant from NIHR (301621). The funders had no role in the study design, data collection, analysis, interpretation, manuscript preparation, or the decision to publish. The correct order of listing is: Younger Family Foundation (YFF) is of the first; and National Institute for Health and Care Research (NIHR) is the next one. We can confirm that the funders played no role in the study design, data collection and analysis, decision to publish, or preparation of the manuscript.

**Competing interests:** The authors have declared that no competing interests exist.

## Introduction

Equitable access to healthcare is a core goal of global health systems and a priority under the Sustainable Development Goals (SDGs), particularly the achievement of Universal Health Coverage (UHC) [1,2]. However, persons with disabilities, who constitute approximately 1.3 billion people worldwide, continue to face significant disparities in accessing quality healthcare services [3]. These inequalities are widespread in low- and middle-income countries (LMICs) like Ethiopia, where health systems are often underfunded and ill-equipped to accommodate the healthcare needs (e.g., general, disability specific) of this population [4]. Consequently, people with disabilities are frequently excluded from essential healthcare (e.g., preventive, promotive, curative or rehabilitation) services, often experience worse health outcomes, reduced quality of life and a shorter life expectancy [3,5–7].

The disadvantage people with disabilities experience in accessing healthcare is caused by multifaceted barriers at both systemic and service delivery levels, ultimately rooted in societal and structural inequities [3,8,9]. Service delivery barriers include limited autonomy, unaffordable healthcare and transport, inaccessible infrastructure, negative attitudes of healthcare workers, lack of disability specific training among healthcare workers, inaccessible information and communication challenges and lack of disability-related services [3,5,9], while health system barriers include poor data systems, shortage of funding for disability inclusion, and inclusive policy gaps [3,5,10–13]. These barriers not only violate the principles of equality and non-discrimination outlined in the United Nations Convention on the Rights of Persons with Disabilities (UNCRPD) [14], but also perpetuate cycles of marginalization and vulnerability [8].

Recent evidence indicates that healthcare barriers are intensified when disability intersects with other forms of marginalization, particularly gender. Women with disabilities face compounded challenges, including poverty, stigma, limited autonomy in reproductive health, and reduced access to screening and diagnostic services [3,9,15]. These inequities contribute to poorer health outcomes and increased mortality risk for women with disabilities [16,17]. The health justice framework asserts that disability is a socio-political identity intertwined with factors such as ethnicity, gender, class, and geography, shaped by systems of marginalization. Addressing it therefore requires systemic and structural change, not merely individual accommodations [18].

Healthcare systems therefore must be strengthened to address the needs of people with disabilities, reduce health disparities, and close the life expectancy gap. These efforts require the removal of healthcare access barriers, as explicitly mandated by the UNCRPD [14] and the African Disability Protocol [19], with member states committed to creating and ensuring an equitable and effective healthcare system. It is therefore crucial to gain a complete picture of what supports or hinders people with disabilities from accessing both general and disability-related (e.g., rehabilitation) healthcare in resource-limited settings. However, studies focusing on the intersection of disability and healthcare access are scarce in sub-Saharan, particularly in Ethiopia [20]. This study therefore aims to explore the barriers and facilitators

of healthcare access for people with disabilities in Ethiopia and to propose solutions for making health systems more disability-inclusive. This study seeks to fill existing evidence gaps by exploring access to both general healthcare (e.g., vaccination, sexual and reproductive healthcare) and disability-related care (e.g., rehabilitation and assistive technology) using a rights-based lens. Moreover, it goes beyond describing barriers, to also identifying facilitators and potential solutions to improving access to healthcare for people with disabilities, thus further expanding the evidence base.

### Ethiopia's healthcare system delivery

Ethiopia is, a low-income country in the Horn of Africa with a population of over 126 million as of 2024 [21]. It operates a primarily public, primary healthcare–oriented system [22,23]. Although there has been progress, health financing remains limited, with only 3–5% of GDP allocated to health—below the global average [24].

The healthcare system functions under a decentralized three-tiered structure, comprising primary (health posts, health centres, primary hospitals), secondary (general hospitals), and tertiary levels (specialized and teaching hospitals) [23]. Public-private partnerships have played an essential role in expanding healthcare coverage, particularly in underserved areas, through initiatives like the Private Health Sector Program [25]. Marginalized groups, such people with disabilities, continue to face significant barriers to care, as equity-focused policies and programmes, like the Health Sector Development Program and the Health Sector Transformation Plan, are often undermined by weak implementation and resource constraints [3,8,23]. These challenges for people with disabilities result from multiple barriers, including physically inaccessible facilities (e.g., lack of ramps and accessible toilets) [11,12], limited training of healthcare workers in disability-inclusive care [11] and high out-of-pocket expenses [22]. Policy dialogues have also aimed to enhance private sector engagement in tertiary care to improve equity and service delivery [6]. However, challenges in ensuring equitable access persist. These inequalities highlight the need for investment in both public and private healthcare sector to deliver inclusive and accessible services for people with disabilities in Ethiopia.

## Methods and materials

### Ethics statement

This study received ethical approval from the London School of Hygiene & Tropical Medicine (LSHTM Ethics Ref: 30698) and Bahir Dar University (BDU Ethics Ref: 1069/2024). Support for data collection was granted by the Amhara Public Health Institute and the city health department. Informed consent was obtained from all participants, with adjustment for those unable to read or write—including thumbprint-witnessed consent for participants with visual impairments and consent from legal representatives for individuals with cognitive impairments. Moreover, reasonable accommodations were provided to ensure inclusive participation of people with disabilities, including communication support (e.g., sign language interpreters), physical accessibility, flexible interview scheduling, involvement of caregivers where appropriate, and adaptation of data collection tools. These measures promoted full engagement and accurate representation of participants' experiences.

Participation was voluntary, and all data were anonymized in accordance with the Helsinki Declaration. Participants were compensated for their time and travel expenses.

### Theoretical framework

This study adopts the Missing Billion Health System Framework—a model specifically designed to systematically assess the inclusion of people with disabilities from health systems [5,9,26] (Fig 1). The framework encompasses four dimensions: system (leadership, governance, financing, and evidence), service delivery (demand-side: autonomy and awareness, affordability; supply-side: healthcare workforce, health facilities, rehabilitation and specialist services), output (effective service coverage), and outcome (health status and responsiveness), with 34 validated indicators for assessing health system performance and 14 for the output/outcome components [26].

**Fig 1. The Missing Billion health system framework.**

In this qualitative study, we focus on the two core dimensions: service delivery (demand and supply) and system-level (financing and evidence) factors. On the demand side, the framework considers the presence of barriers such as stigma, limited autonomy in decision making, and lack of accessible health information that shape individuals' ability to seek and engage with care [9]. The supply side addresses the readiness of health systems to provide inclusive services, including infrastructure accessibility, provider capacity, and respectful communication [5,9,26]. In the Ethiopian context—marked by poverty, infrastructural barriers, and limited implementation of inclusive health system policies—the framework offers a practical and disability-specific lens to identify actionable strategies for strengthening health systems where people with disabilities are expected, accepted, and connected [5,27].

## Study design

An exploratory qualitative study was conducted in Bahir Dar City, Ethiopia in November and December 2024 [28] to investigate barriers and facilitators to accessing general and disability-related healthcare.

## Study setting

The study was conducted in six sub-cities of Bahir Dar, the capital of Amhara Regional State (Fig 2). Bahir Dar, the country's fifth-largest city, has an estimated population of 249,851 and lies on the southern shore of Lake Tana [29]. In 2010, the disability prevalence in the region was 14% [30]. By 2022, the city's Department of Women, Children, and Social Affairs reported around 2,098 individuals with various disabilities, including mobility, hearing, visual, intellectual impairments, and leprosy. The city hosts one assistive technology workshop, a physical rehabilitation centre, and several governmental and non-governmental organizations advocating for disability rights and healthcare access. Its healthcare infrastructure includes ten primary health facilities, four public hospitals, four private hospitals, and numerous clinics, drug stores, and diagnostic laboratories.

## Study participants and sampling

Adults with disabilities living in Bahir Dar City were recruited using purposive sampling followed by snowball sampling. Initially, participants were identified with support from Organizations of Persons with Disabilities (OPDs). Subsequent participants were referred by those already interviewed. From six sub-cities, 30 adults were purposively selected out of 36

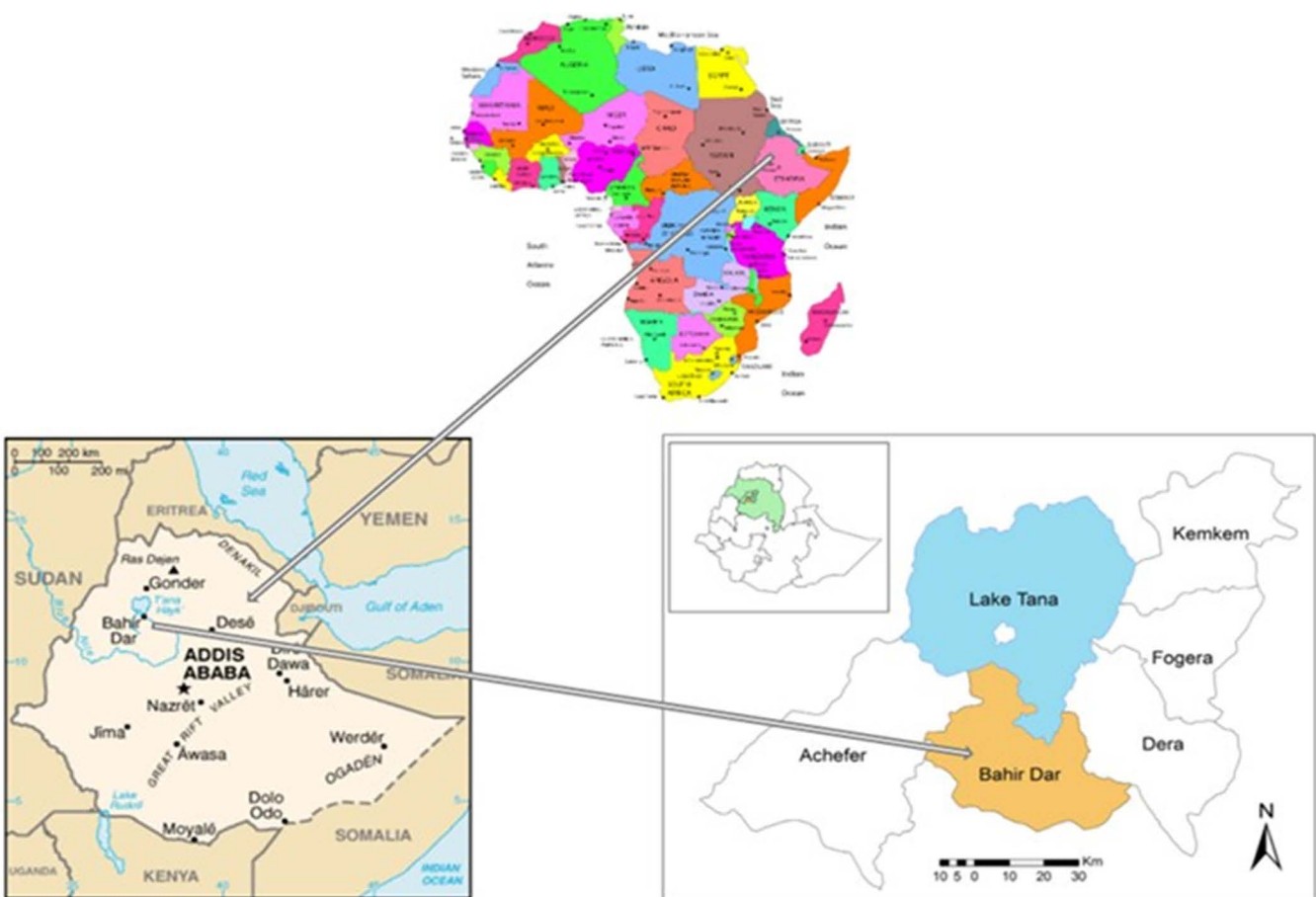

**Fig 2. Map of Bahir Dar City.** Source: Natural Earth. Free vector and raster map data @ naturalearthdata.com. Public domain.

contacted individuals, ensuring a diverse range of perspectives. The focus was on adults, as in contrast to children, they typically make independent healthcare decisions and can describe their own experiences. All participants self-identified as having a disability and were screened using the Washington Group Short Set questions, with inclusion limited to those reported "a lot of difficulty" or more in at least one domain. Maximum variation sampling ensured diversity across age, sex, sub-city, and impairment type [31]. Despite efforts to include individuals with intellectual disabilities directly, caregivers were interviewed on their behalf due to challenges understanding and communicating. The final sample included five participants with cognitive/ intellectual impairments, six who were deaf or hard of hearing, eight had a visual impairment, and eleven had a physical impairment. Recruitment continued until data saturation was achieved [31].

## Data collection

The research team developed a semi-structured interview guide (S1 Text) to explore the healthcare access experiences of people with disabilities and their caregivers. The tool covered socio-demographic characteristics, healthcare knowledge and attitudes, affordability, service availability, provider interactions, transportation, physical environment, assistive technology, and policy-related issues. It was pilot-tested with three individuals with visual, hearing, and physical impairments to refine the approach. In-depth interviews were conducted by DDA in Amharic, the local language spoken in Ethiopia. We applied a probing technique using follow-up questions until data saturation was reached. Interviews took place at safe,

participant-chosen locations (homes, workplaces, or public spaces), ensuring privacy and comfort, or, if preferred, online. Informed consent was obtained, and sessions were audio-recorded.

Reasonable accommodations included sign language interpretation for participants with hearing impairments, and involvement of caregivers in interview with persons with cognitive impairments. Interviews lasted between 32 and 154 minutes, with an average of 102 minutes per interview. The team (DD, HK, and FBM) held regular meetings to reflect on sampling, the interview process and emerging findings.

### Data management and analysis

Interview recordings were transcribed verbatim into Amharic and then translated into English by DDA and an independent transcriber, or directly transcribed into English. Each transcript was checked against the audio recordings for consistency. All transcripts were de-identified and securely stored on a protected main drive. Transcripts were exported into NVivo 14 for coding and organization. Data was coded using deductive coding with a predetermined codebook (S1 Table) developed from the interview guide and piloted transcripts, based on the Missing Billion framework, and subsequently refined to incorporate emerging codes and themes [32]. The thematic analysis followed six steps [33]: (1) familiarization with the data, (2) line-by-line coding in NVivo 14 using the codebook, (3) grouping codes into thematic charts and exploring relationships, (4) reviewing and refining sub-themes and themes, (5) defining and naming themes against transcripts and codes, and (6) extracting narratives and quotes to support findings. Findings were reported according to the consolidated criteria for reporting qualitative research (COREQ) [34]. The analysis, structured within Missing Billion Health System Framework, captured both demand- and supply-side factors influencing healthcare access [26]. Finally, to ensure rigor, we engaged in prolonged involvement with participants, maintained researcher reflexivity, and conducted peer debriefing and inter-coder reliability checks. We provided rich contextual descriptions and adhered to COREQ guidelines to enhance transparency. Diverse sampling strategies, including maximum variation and snowball sampling, supported credibility, transferability, dependability, and confirmability.

### Researcher characteristics and reflexivity

The research team comprised DDA, a PhD student with a disability and lived experience navigating Ethiopia's health and social systems, and HK and FBM, experienced PhD supervisors with expertise in disability research and qualitative methods in sub-Saharan Africa. Each brought unique perspectives that enriched depth and rigor of the study. DDA's positionality as a person with a disability facilitated empathetic engagement and trust with participants, offering deeper insights into healthcare access barriers and facilitators. Due to gender dynamics between a male interviewer and female participant, particular care was taken when discussing sensitive topics. Participants were reminded of their right to skip questions or stop the interview at any time to ensure voluntary and comfortable disclosure. Reflexive journals, regular peer meetings, and collaboratively developing codes and themes through iterative discussions enhanced the integration of lived experiences, credibility and authenticity. We remained aware of the potential for recall bias, particularly in narratives involving long-term or repeated healthcare interactions. To minimize recall bias, participants were encouraged to share recent or concrete experiences whenever possible, and follow-up questions were used to verify or clarify timeline and context.

## Results

Thirty participants and their caregivers completed interviews across six sub-cities in Bahir Dar City. Most participants were male, average age was 34 years (range: 18–57 years) and they have lived with a disability for 27 years on average (range 8–51 years). Disabilities were categorized into physical (11), intellectual (5), and sensory (hearing (6) and vision (8)) impairments. Few participants have enrolled into community-based health insurance, and most were both married, and employed. Health literacy varied, with some undergoing regular check-ups while others relied on traditional remedies. Initial coding of interview transcripts identified 82 nodes (26 barriers, 31 facilitators, and 25 coping strategies).

After refinement, five primary themes emerged for both barriers and facilitators—further categorized by demand- and supply-side aspects of service delivery—along with five distinct themes related to coping strategies, each encompassing multiple sub-themes influencing access to healthcare (– S2 Table).

## A. Barriers to access primary healthcare

**Demand-side barriers.**

**Theme I: Autonomy and awareness**   Participants highlighted that lack of knowledge and low literacy limited their ability to recognize health needs, understand service availability, and make timely healthcare decisions, partly due to exclusion from the education system.

> *"… Because health is tied to education, many are excluded from the education system, they are illiterate like me…"* (Male with physical impairment)

Many delayed seeking care, especially for seemingly minor symptoms. A lack of information about specialized services worsened the situation, often perceived to lead to poorer health outcomes. Misinformation rooted in religious beliefs reduced the ability of people with disabilities to recognize their health conditions and seek appropriate care. In particular, religious beliefs that regard disability as unmanageable with clinical treatment methods and associated it with spiritual causes further discouraged the use of available medical services.

> *"…Communities believe disability-related issues cannot be treated medically and prefer religious practices instead. People with disabilities are often seen as incurable..."* (Male participant with physical impairment)

Additionally, anxiety and fear—particularly associated with prior negative encounters—were important deterrents to timely healthcare utilization. A pregnant woman who requested priority service during an ANC visit for universal care of the foetus and maternal health described experiencing humiliation:

> *"…During my pregnancy, I requested priority at an ANC visit but was met with humiliating remarks from provider — saying you are pregnant in your uterus, not in your eyes. This response deeply embarrassed me and, despite asking as such is my right, I became too afraid to seek further assistance afterward..."* (Female with vision impairment).

Family and community support also emerged as critical. Some participants reported neglect or active discrimination within their own households, which further isolated them and made it harder to reach health services independently. These challenges were especially reported by women with disabilities, who faced both gender- and disability-based exclusion and frequently said they had limited power in making decisions about their health. For example, a participant described:

> *"…Though providers said my right eye could improve if I were seen early, I lacked timely treatment due to poor family support and finances. During a childhood, my family never took me to a clinic—only to holy water. Disabled children often face unequal healthcare, clothing, and opportunities..."* (Male with visual impairment)

**Theme II: Affordability**   Financial hardship was reported to be a major barrier to healthcare access. People with disabilities often face limited employment opportunities and disability-related needs placed them in persistent economic vulnerability. This economic strain was worsened by added costs of assistive devices, specialized caregivers, and travel expenses to reach accessible health facilities. Out-of-pocket payments continued to dominate health financing in Ethiopia, increasing financial vulnerability and deterring individuals from seeking healthcare services. While health insurance was theoretically available, participants reported inconsistent coverage and unclear

entitlements. Individuals with disabilities were either denied services at the point of care or advised not to rely on insurance, undermining trust in the overall service effectiveness. Inconsistently applied fee waivers and bureaucratic bottlenecks—both within healthcare facilities and government bodies—further discouraged enrolment and usage. Moreover, frequent stockouts of essential medicines and inadequate diagnostic services in public facilities forced many to seek private care, significantly increasing financial burden. This was distressing for those already navigating disability-related costs. As one participant recounted:

*"I wish to visit a good healthcare facility for the eye health issue, but I don't have health insurance, as we were advised against using it. On the other hand, even when health insurance is available, it is often disregarded." (Male with vision impairment)*

Another shared:

*"Essential drugs, especially costly ones, are often out of stock, forcing us to buy from private pharmacies at higher prices." (Female with vision impairment).*

**Supply-side barriers.**
**Theme III: Healthcare workers**   Participants frequently described that healthcare providers lacked both pre-service and in-service training to respond the specific needs of people with disabilities. This gap, coupled with misconceptions of disability, led to poor service quality. For instance, several participants reported that providers expressed surprise when women with disabilities were pregnant or seeking reproductive care—reflecting knowledge deficits.

*"When disabled women seek antenatal care, they hear remarks like, 'You too? You're pregnant?' This reflects both a knowledge gap and lack of ethics. They look down on them, failing to recognize that sexuality is natural. Such comments are hurtful" (Male with physical impairment)*

Moreover, participants emphasized that the lack of professional training compromised care quality and increased their exposure to disrespect and discrimination. Many healthcare providers were also uninformed of national policies protecting the rights of people with disabilities, often relying on personal discretion rather than upholding their professional obligations.

Beyond knowledge gaps, participants frequently encountered dismissive attitudes and disrespect from health workers, including offensive language, which made them feel unwelcome in healthcare spaces and discouraged care-seeking. These negative interactions had emotional impacts, and also delayed access to timely and appropriate care. For example, a female participants said:

*"We feel that healthcare professionals do not treat us with the same respect as other patients. For these reasons, many disabled individuals avoid seeking healthcare, even during illness." (Famale with vision impairment)*

Participants also raised concerns about service quality, citing breaches of confidentiality due to third-party assistants, poor communication, and a lack of reasonable accommodations. For instance, a participant described:

*"Masks have become a major barrier for us. While essential for health, they make communication very difficult for deaf people like me." (Female with hearing impairment)*

Others mentioned that their input was often disregarded, and providers failed to involve patients in decision-making when prescribing medications—an especially pressing concern for economically disadvantaged individuals with disabilities.

Additionally, timely access to services remained uncertain even after arrival. Participants explained that, despite their mobility challenges, they often wait in long queues without accommodation. Health professionals rarely prioritized their needs, leading to delays and feelings of neglect, as explained by the respondent below.

*"This is one of the challenges I faced. After waiting for six hours, they told us they were going to have lunch, despite many patients waiting for treatment. I patiently waited in line, but when they returned, they said my results hadn't arrived from the laboratory. Afterwards, the doctor then blamed me for the delay and told me to fix the issue."* (Male with hearing limitation)

**Theme IV: Healthcare facilities** Participants consistently reported structural accessibility barriers. Health centres lacked basic accessibility features —such as ramps, wide doorways, and adapted examination rooms. Even where some improvements had been reported, the lack of adaptive equipment like adjustable beds or accessible service windows continued to limit appropriate care. For example, for wheelchair users, even reaching a service window could be a challenge. These barriers had potentially serious consequences, particularly in emergencies, as illustrated by a participant who described a pregnant woman's struggle with facility inaccessibility during delivery.

*"I know a pregnant woman using a wheelchair was unable to enter the delivery ward due to her wheelchair not fitting through the door. She faced discrimination, was told to crawl, and struggled to get onto the delivery bed as no staff offered any assistance."* (Female with hearing impairment)

Participants also mentioned transportation challenges further exacerbated access difficulties. Public transport systems often lacked inclusive design and treated people with disabilities unfairly.

*"There is no transportation system specifically designed to consider us. Specially, loading and unloading the wheelchair is challenging, and the vehicle is too high to climb for those who use crutches."* (Man with vision impairment)

Navigating the environment to reach health facilities posed additional risks—particularly for those with visual or mobility impairments, as one participant described:

*"I may not notice a newly built canal, but poles, ditches, and other obstacles often block walkways, increasing the risk of collisions or falling into deep holes."* (Female with vision impairment)

The health system showed a critical lack of responsiveness in communication strategies, often overlooking the needs of people with disabilities. Key health campaigns—on immunization, maternal care, and disease prevention—were reported to rarely reach those with hearing, visual, or cognitive impairments. This gap led to missed care opportunities and reinforced feelings of exclusion. Participants voiced strong concern about the absence of accessible health information in formats that accommodate diverse communication needs, leaving many uninformed about essential services and campaigns. One participant highlighted how mainstream health communication failed to adapt to diverse needs:

*"…Healthcare information in Bahir Dar, Ethiopia, is not accessible in various formats for different disability types. For example, deaf individuals need sign language interpretation or captions to access television broadcasts for health information…"* (Male with physical impairment)

**Theme V: Rehabilitation, specialized services and AT** Many participants explained that rehabilitation, specialist services and AT were either unavailable or poorly integrated into routine healthcare. Besides, outreach efforts rarely responded to their needs, with few home visits or targeted screenings. AT—crucial for independent and safe access to care—was often unavailable. As one participant reported:

*"… campaigns seldom focus on people with disabilities or conduct door-to-door visits to identify those who may need disability-related healthcare… Supportive assistive devices are essential, yet rarely available." (Male with vision impairment)*

## B. Facilitators to access primary healthcare

**Demand side facilitators.**

**Theme I: Autonomy and awareness** A few participants, empowered by personal knowledge and experience, confidently navigated the health system, engaged with providers, and practiced preventive care—demonstrating the value of education in building self-advocacy skills.

*"We can prevent diseases by practicing preventive behaviours, seeking advice from healthcare professionals, and applying their recommendations." (Male with vision impairment)*

Family members played a vital role in facilitating access, including communication support, financial assistance, and accompaniment during visits. Community networks, peer and disability associations also offered practical support, encouragement, and health education.

*" My friends and family support me by providing moral support, transportation, and assistance within the facility to help me receive better services." (Male with vision impairment)*

**Theme II: Affordability** Health insurance was described as important to reduce financial burdens of people with disabilities, facilitating access to services, medications, and assistive technologies. Some also benefited from financial support by disability-focused organizations, which subsidized premiums. Fee waiver program also offered financial protection, though the application process was often difficult, as explained by some respondents.

*"Health insurance now covers his costly epileptic medication. Without it, the expense would be overwhelming, especially as a mother I am supporting other children." (Caregiver, female, son with cognitive impairment)*

**Supply side facilitators.**

**Theme III: Healthcare workers** Disability-specific training was perceived to improve providers' awareness and responsiveness, fostering respectful and inclusive care. Respondents explained that in some facilities, NGO-trained staff were better equipped to deliver services using sign language and disability-sensitive approaches.

*"If well-trained professionals, including social workers fluent in sign language and those proficient in Braille, are available, people with disabilities could access healthcare equally." (Male with vision impairment)*

Some participants described how supportive attitudes—particularly from providers with personal ties to disability—enhanced their care experiences. Empathy and proactive support fostered participants to feel respected and valued by

health system. Others emphasized that providers' ethical commitment—shaped by personal values, faith, or professional oaths — built trust and encouraged continued care-seeking.

Respectful, effective communication was vital for deaf or hard-of-hearing participants. While formal sign language services were rare, some providers used writing or gestures to support understanding and preserve patient autonomy.

*"They communicate with me through lip reading and written notes… This works for me, but it is difficult for others."* (Male with hearing impairment)

Some public health facilities in Bahir Dar City clearly prioritized people with disabilities by providing reasonable accommodations in line with disability rights frameworks.

**Theme IV: Healthcare facilities** Consistent medication availability in public facilities helped prevent treatment interruptions and reduced associated costs, supporting financial stability. So, subsidized services in public facilities further improved treatment adherence among individuals with disabilities, who are more likely experiencing poverty. For example, one respondent explained:

*"The service is now improved with well-supported medical equipment and skilled professionals… Diagnosis and treatment are effective, and I am fully satisfied."* (Male with vision impairment)

Several respondents highlighted improvements in the physical accessibility of newly constructed or renovated facilities, reducing reliance on caregivers and enhanced independent access to healthcare.

*"Recently, accessibility in newly built healthcare facilities has improved… Two health centres in the city have undergone infrastructure innovations as model facilities for disability inclusion."* (Male with vision impairment)

Media and digital platforms were also mentioned as important tools in raising awareness and empowering people with disabilities to seek care, especially when formats were accessible and included disability-relevant content.

*"I obtain information from social media, mainstream media, the internet, other people's experiences, and the WHO website."* (Male with vision impairment)

**Theme V: Specialized services & AT** Support from NGOs and the Bahir Dar Rehabilitation Centre—through free or subsidized devices, maintenance, and medications—was reported which able to reduce financial burdens. Public servant respondents with disabilities also noted receiving government support for AT.

*"With Cheshire's support, I accessed continuous treatment. The Bahir Dar Rehabilitation Centre provided canes at low or no cost, and as a public servant, the government covered cost of my assistive devices."* (Male with vision impairment)

## C. Solutions to overcome barriers to access primary healthcare

While healthcare access is a challenge for everyone, people with disabilities experience additional, disability-specific barriers that demand tailored, inclusive responses. Solutions must be designed with their unique barriers in mind, ensuring they are relevant and responsive to the lived realities of people with disabilities. Accordingly to the framework, the following solutions were suggested by participants.

**Theme I: Autonomy and awareness.  Public education and media awareness campaigns.** Media campaigns can powerfully challenge stigma and promote disability inclusion. Participants highlighted the need for nationwide efforts—especially in rural areas—to shift public attitudes and drive inclusive policy change.

*"The media should raise awareness that disability can affect anyone and that all disabled individuals need community and family support. This would promote equal treatment, reducing confusion and stigma, especially for those with intellectual disabilities." (Male caregiver, daughter with intellectual impairment)*

**Theme II: Affordability.  Enhance inclusive budget allocation and healthcare financing.** Participants widely recognized the link between poverty and limited healthcare access. There were repeated calls for greater budget allocation, better utilization of existing funds, and the development of inclusive financing mechanisms to reduce out-of-pocket expenses and improve health outcomes.

*"They should also allocate enough funding to subsidize the medical supply needs of people with disabilities." (Male with physical impairment)*

Moreover, disability budgets already allocated need to be properly utilised in a transparent way to achieve the intended impact.

**Village based funding.** Given the limitation of out-of-pocket payment system in Ethiopia, participants recommended community-based funding initiatives, like village savings and loans schemes, to support the purchase of essential assistive technologies such as wheelchairs.

**Increase insurance coverage.** Participants strongly suggested targeted government subsidies to enable people with disabilities enrol in health insurance scheme. Insurance was identified as a promising strategy to reduce direct costs and increase access to services.

**Theme III: Healthcare workers.  Disability training for healthcare workers.** Several participants emphasized the need to train healthcare professionals to improve disability awareness, challenge biases, and ensure equitable care, and uphold legal and ethical obligations.

*"All stakeholders, including policymakers, advocacy groups, and healthcare institutions, must prioritize disability awareness to ensure equitable and respectful service provision." (Female with vision impairment)*

Sustainable service improvements depend on integrating disability into medical education and policy frameworks.

**Assigning personal assistance in healthcare facilities.** There was a strong call for the deployment of personal assistants in healthcare centres to support them in mobility, communication, and care navigation— thereby enhancing dignity and promoting inclusive, responsive care.

**Theme IV: Healthcare facilities.  Action to enhance accessibility.** Most participants stressed that healthcare facilities must be adapted to meet the physical access needs of people with disabilities.

*"Healthcare facilities must be made accessible to ensure wheelchair users can receive medical services without difficulty... No one should have to crawl to access medical services—facilities must be designed to ensure dignity and accessibility for all." (Female with physical impairment)*

**Ensuring supply of medical equipment and medication.** Some participants linked the availability of medical supplies to improve service quality, affordability, and satisfaction, particularly for those insured disabled people.

**Ensuring accessible health information and reasonable adjustment.** All participants advocated providing health information in accessible formats—such as sign language, braille, audio, and easy-to-read materials—and stressed the need for functional information centres.

**Establishing disability units and inclusive governance structures.** Most participants emphasized the need to mainstream disability throughout the health system by establishing dedicated units that coordinate disability issues. They also stressed the need to ensure meaningful participation of people with disabilities in policy and decision-making. They argued that responsive, rights-based solutions can only emerge through the involvement of those with lived experience.

   **Theme V: Rehabilitation, AT and specialized services. Improve availability and affordability of AT.** Most participants emphasized the critical role of assistive devices like wheelchairs in enabling access to healthcare. However, affordability and misuse were raised as key issues. Suggestions included promoting local production and ensuring proper distribution and monitoring systems. Expansion of disability-related healthcare (e.g., rehabilitation) also commendable.

   *"I believe wheelchairs can be best manufactured locally in our country... If assistive technology can be produced locally, most individuals with disabilities would be able to afford them." (Male with physical impairment)*

**Summary of key results (Fig 3)**

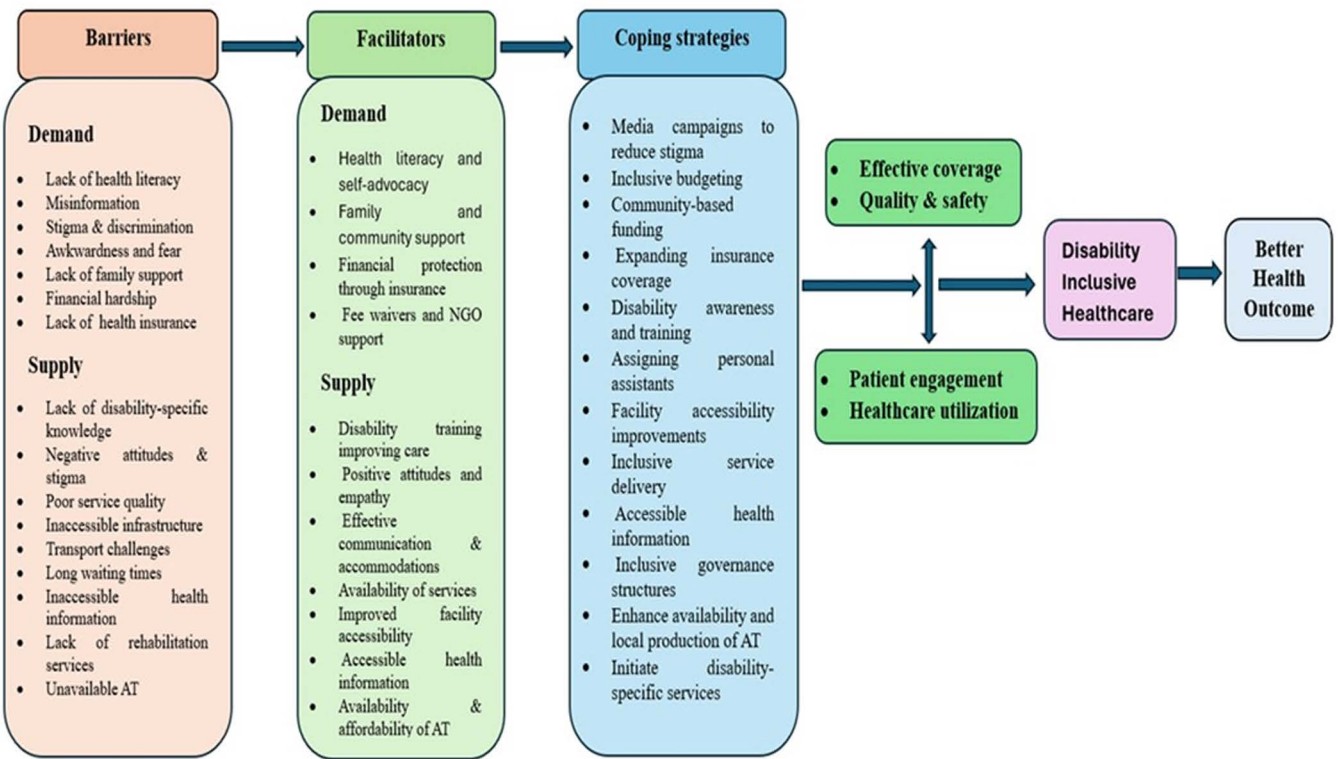

**Fig 3. Summary of key results.**

## Discussion

Despite Ethiopia's commitments to disability rights—evident in its ratification of the UNCRPD [15] and adoption of a national disability proclamations [35]—this study found that people with disabilities report multiple barriers when accessing both general and disability-related health services, adding to the sparse literature on this topic. However, participants also identified facilitators and potential solutions that have and could improve access to healthcare, which has rarely been explored in a sub-Saharan African context. These barriers, facilitators, and solutions were systematically mapped onto five core categories across demand- and supply-side domains using the Missing Billion Health System framework. While aligning with existing global and national evidence, this study provides new, context-specific insights, emphasising the urgent need for Ethiopia to translate its disability rights commitments into inclusive health system reforms that guarantee equity, access, and dignity.

Demand-side barriers included low health literacy, misinformation, stigma, fear, lack of family support, financial hardship (e.g., high costs of transportation, medication, and assistive devices), and limited health insurance mirroring patterns reported in the sparse Ethiopian literature [11,12,22,36–39] and from international studies [40,41]. Also consistent with prior findings [3,22,39], this study demonstrated how poverty and disability intersect to delay care-seeking. This pattern appeared to be intensified by Ethiopia's heavy reliance on out-of-pocket health spending (31% of expenditures) and inconsistently implemented fee waiver schemes [22], leaving 95% of people with disabilities reporting to be unable to afford healthcare [42]. This trend is also corroborated by evidence from other LMICs, where financial insecurity remains a principal barrier to healthcare for people with disabilities [40,43–45]. Additionally, chronic underfunding has further limited service coverage, and quality of care [22,23,46]—factors widely reported by participants with disabilities as reasons for low healthcare utilization. Such persistent disparities in coverage and access jeopardize progress toward the SDGs [41].

Supply-side barriers identified in this study included negative provider attitudes, poor service quality, transport challenges, and lack of rehabilitation services and were also consistent with national [11,12,37,39] and global reports [3,4,10,13,40]. Our study found that discriminatory practices and poor communication skills among providers—resulted from limited disability knowledge and inadequate disability training—further marginalized people with disabilities, again supported by findings from other LMICs [4,40,43,44,46,47]. Moreover, the study also reported discriminatory remarks and poor communication in reproductive health settings, often due to provider's lack of sign language skills and awareness of inclusive practices, further corroborated by a previous study [37]. These challenges reflect broader gaps in both pre-service and in-service disability training [3].

Structural inaccessibility (e.g., missing ramps, narrow doorways, inaccessible examination tables, and lack of service adaptations) violate legal obligations under the UNCRPD [15] and national disability mandates [35,48,49], severely compromising access to healthcare, as also shown by previous studies [4,44,46,47]. These persistent structural barriers, despite some minor reported improvements, continue to breach global and national obligations [14,15,35,42,48,49]. Although accessibility is a legal and ethical imperative under Primary Health Care, UHC, and the SDGs [50], implementation remains inconsistent. The absence of accessible equipment, such as adjustable beds and examination tables [44, 45], highlighted systemic gaps and reinforces the urgent need to embed a rights-based approach into Ethiopia's health system to ensure dignity, non-discrimination, and equity [14,15,38].

Some positive experiences—such as trained staff using sign language, supportive provider attitudes, and prioritized care in Bahir Dar healthcare facilities—contrast with earlier studies that reported widespread neglect and discriminatory behaviours [47,51]. These localized improvements suggest early gains from targeted disability inclusion efforts [38]. Likewise, the use of adapted communication methods, such as written text or gesture, and the provision of fee waivers aligns with best practices seen in other LMICs [38]. However, bureaucratic challenges continue to hinder effective access to the waiver system [11,52]. This reflects broader patterns in low-income settings, where well-intentioned policies often fail in practice due to administrative inefficiencies [11,47]. Streamlining access through community-based identification or digital platforms may improve efficiency and reach, especially in rural areas.

Notably, the call for community-based financing mechanisms and expanded insurance coverage confirms global evidence linking disability, poverty, and financial barriers to healthcare [47,52]. Furthermore, AT products are typically scarce and imported in LMICs [44,45]. The emphasis on affordable, locally produced AT presents a context-specific alternative to costly, import-dependent models commonly described in prior studies [38,45,53]. Participants also proposed more systemic, rights-based solutions, including integrating disability into medical curricula, deploying personal assistants in health facilities, and establishing dedicated disability units within health systems — suggestions rarely highlighted in earlier researches [11,37]. These actionable proposals directly align with the UNCRPD's calls for inclusive governance and participatory health systems [15], reinforcing broader shifts in global disability inclusion thinking [15,38].

## Strengths and limitations

This study's strength lies in its use of a rights-based, systems-level framework (the Missing Billion Health System framework) and rich firsthand accounts from people with diverse impairments. Data were collected by a trained researcher with disabilities in the local language, enhancing trust and contextual insight. We also emphasis on moving beyond barriers to uncover potential facilitators and recommend practical solutions. The effort of the study in the urban area may limit transferability to rural or pastoralist where barriers could be more severe. This study might also be subject to recall and social desirability bias, as participants were asked to reflect on past experiences with healthcare access. Gender dynamics—particularly the male interviewer's role—might have influenced participant disclosure, especially among women discussing sensitive health topics. The overlaps between framework components—such as affordability (service delivery) and healthcare financing (system-level)—blurred distinctions between individual cost burdens and system resource constraints, complicating coding, analysis, and designing targeted interventions. Despite scholars call for decolonial, intersectional approaches that reflect the lived realities of marginalized groups, especially women with disabilities and those in informal urban settings, this study gave limited attention to how gender, poverty, and urban marginalization intersect to shape healthcare access, restricting the study analytical depth. Additionally, ambiguity between general and disability-related care complicated interpretation, as participants often discussed systemic issues—like drug shortages or insurance gaps—without clearly linking them to the type of service required. In Ethiopia, where socioeconomic inequality, gender norms, and rapid urbanization intersect with under-resourced disability services converge, this gap is particularly relevant. The framework also did not fully consider characteristics of respondents or capture intersectional factors such as gender, education and poverty. Following strict bracketing was challenging, consistent with interpretive phenomenology's recognition of the researcher's role in co-constructing meaning. The absence of provider or policymaker perspectives limited triangulation, might leave some system-level constraints and policy feasibility unexplored. Future studies should address these limitations through mixed-methods, longitudinal designs, broader sampling strategies and a wider geographical scope.

## Implications and policy practices

Addressing healthcare barriers for people with disabilities in Ethiopia requires systemic reforms targeting accessibility, affordability, provider training, autonomy and inclusive governance. Policies should prioritize disability-specific training, inclusive health financing, assistive technology access, and infrastructure adaptation. Participants' coping strategies—advocating for dedicated budgets, community financing, and inclusive governance—highlight practical, experience-based solutions. Strengthening community support and integrating disability into national health strategies are vital for advancing equitable, rights-based healthcare. Translating these insights into participatory, action-driven reforms is essential to achieving universal health coverage and fulfilling the commitments of the UNCRPD.

## Conclusion

This study revealed persistent barriers to healthcare access for people with disabilities in Ethiopia, alongside promising facilitators and community-driven coping strategies. Addressing these challenges requires sustained policy commitment,

coordination, and the meaningful integration of disability perspectives within routine healthcare delivery system. Amplifying the voices of people with disabilities and embedding rights-based, participatory approaches are essential to advancing universal health coverage and achieving equitable care for all. These efforts directly support the achievement of Sustainable Development Goals (SDGs), particularly SDG 3 (Good Health and Well-being) and SDG 10 (Reduced Inequalities), by ensuring that no one—especially vulnerable populations with disabilities—is left behind.

## Supporting information

**S1 Text. Interview guide.**
(DOCX)

**S1 Table. Codebook.**
(DOCX)

**S2 Table. Summary of themes mapping.**
(DOCX)

## Acknowledgments

We sincerely appreciate all the participants who dedicated their time and participation in our study. Our special thanks go to the Bahir Dar City Health Department and the City Association of People with Disabilities, with particular recognition to Mr. Firie Selam. We are also grateful to the Disability Research Group in Entebbe, Uganda, for their valuable guidance on transcript coding and the use of NVivo. We are grateful to Prof. Mezgebu Yitayal for his suggestion on the codebook draft. Finally, we extend my thanks to the Department of Health Systems Management & Health Economics at Bahir Dar University for their support with printing services and stationery supplies.

## Author contributions

**Conceptualization:** Desta Debalkie Atnafu, Hannah Kuper, Femke Bannink Mbazzi.

**Data curation:** Desta Debalkie Atnafu, Hannah Kuper, Femke Bannink Mbazzi.

**Formal analysis:** Desta Debalkie Atnafu, Hannah Kuper, Femke Bannink Mbazzi.

**Investigation:** Desta Debalkie Atnafu, Hannah Kuper, Femke Bannink Mbazzi.

**Methodology:** Desta Debalkie Atnafu, Hannah Kuper, Femke Bannink Mbazzi.

**Project administration:** Desta Debalkie Atnafu, Hannah Kuper, Femke Bannink Mbazzi.

**Software:** Desta Debalkie Atnafu.

**Supervision:** Hannah Kuper, Femke Bannink Mbazzi.

**Validation:** Desta Debalkie Atnafu, Hannah Kuper, Femke Bannink Mbazzi.

**Writing – original draft:** Desta Debalkie Atnafu.

**Writing – review & editing:** Desta Debalkie Atnafu, Hannah Kuper, Femke Bannink Mbazzi.

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
