## [Decision Letter · Decision Letter 0]

9 Jul 2025

PGPH-D-25-01404

Left behind in primary healthcare: A qualitative exploration of healthcare experiences of people with disabilities in Ethiopia

Dear Dr. Debalkie,

Thank you for submitting your manuscript to PLOS Global Public Health. After careful consideration, we feel that it has merit but does not fully meet PLOS Global Public Health’s publication criteria as it currently stands. Therefore, we invite you to submit a revised version of the manuscript that addresses the points raised during the review process.

We look forward to receiving your revised manuscript.

Kind regards,

Damen Haile Mariam, MD, MPH, PhD

Academic Editor

Journal Requirements:

3. In this instance it seems there may be acceptable restrictions in place that prevent the public sharing of your minimal data. However, in line with our goal of ensuring long-term data availability to all interested researchers, PLOS’ Data Policy states that authors cannot be the sole named individuals responsible for ensuring data access (http://journals.plos.org/plosone/s/data-availability#loc-acceptable-data-sharing-methods).

Additional Editor Comments (if provided):

Reviewer 1:

Overall Assessment and Recommendation:

- This manuscript addresses a critical and underrepresented issue in global health—the healthcare experiences of people with disabilities in Ethiopia—and offers rich, participant-centered insights grounded in a rights-based framework. The topic is timely, the methodology is generally sound, and the findings have strong potential to inform policy and practice.

- However, the manuscript would benefit from clearer articulation of its unique contribution to existing literature, a more precise methodological framing, improved findings and discussion sections, and discussion of limitations outlined below. With thoughtful revision, this study has the potential to make meaningful contribution to PLOS Global Public Health and to the broader discourse on disability-inclusive health systems.

Introduction:

- The introduction section is well-structured with appropriate references to disability rights and access literature.

- However, it would have benefited from employing disability justice or feminist disability studies to better surface the intersectional nature of the barriers. This would have enhanced the global relevance of the argument raised in the manuscripts and move it beyond descriptive public health framing.

Theoretical Framework

- The Missing Billion framework seems to be an appropriate one for applied health systems analysis and disability-inclusive design.

- Yet the framework is not rooted in qualitative approach or phenomenology.

- So:

- If the goal is to inductively indicate lived experience, the framework should not be used to the code qualitative data directly.

- It should be introduced in the Methods as a secondary analytic or interpretive tool, not as a deductive framework.

- I recommend the authors clarify how and when the framework was used in the research process. Was it applied after thematically analyze data, to interpret emergent findings, or did it guide interview design and initial coding? If the study is primarily inductive and experience-focused, the framework should be presented as a secondary interpretive lens. This distinction would improve methodological transparency and preserve the integrity of the qualitative approach.

Methods:

- The authors indicated that the study is “phenomenological,” yet the analysis follows reflexive thematic analysis.

- Consider rephrasing this as a qualitative exploratory study informed by phenomenological approach, rather than claiming a strict phenomenological design—since phenomenological analysis techniques such as bracketing, horizonalization, textural and structural synthesis, and essence formulation—are not clearly employed or reported.

- I suggest the authors either:

a) Clarify and justify their use of a phenomenological method by explicitly outlining steps taken (e.g., bracketing, meaning-making, synthesis), or

b) Reframe the methodology as a qualitative thematic study informed by phenomenological sensitivity, rather than a strict phenomenological design. This would better reflect the analytic procedures

and strengthen the study’s methodological coherence.

- The manuscript appropriately highlights the lead researcher's positionality as a person with a disability, which likely facilitated trust and empathetic engagement with participants.

- However, given the gendered dimensions of healthcare access discussed in the findings—particularly by women with disabilities—reflecting on how gendered power relations influence data collection, rapport, and interpretation, would have strengthened the pieces on reflexivity.

Results:

- The presentation of barriers, facilitators, and solutions is rich and logically structured.

- The manuscript includes references to percentages when describing the background of participants. I recommend avoiding quantitative expressions in this context, as they can mislead readers into interpreting the data as statistically representative. It will be better to focus on variation than aggregation. Instead, consider using qualitative descriptors such as “several,” “many,” or “a few.

- On Beliefs in Incurability as a Barrier: The manuscript frames beliefs around the “incurability” of disability as a barrier to care. However, it is essential to distinguish between curability and treatability in the context of disability. Many disabilities are indeed not curable but can be effectively managed with appropriate treatment methods. Therefore, framing the belief in incurability as “misinformation” could undermine realistic understandings of chronic disability conditions. I recommend the authors reframe this point to recognize that while biomedical cures may not exist for all disabilities, this does not equate to hopelessness or justify neglect in care.

- After finishing reading the manuscript, I have still confused about the scope of the healthcare access the authors explored. Several participant quotes indicate both general and disability-specific healthcare, making it difficult to clearly delineate the scope of services under discussion. For example, quotes about drug stockouts, insurance exclusion, or religious interpretations of disability are not consistently framed as applying to all health needs or only to disability-specific care. Clarifying this distinction particularly in the introduction and methods—would help the reader interpret whether the reported barriers reflect exclusion from the broader healthcare system or inadequacies in disability-targeted care. This distinction also has implications for policy and interventions.

- Given the phenomenological and retrospective nature of the in-depth interviews, there is a potential for recall bias, particularly as some participants reflected on healthcare experiences that occurred many years ago—including during childhood. While the study emphasizes rich narrative accounts, it would strengthen the methodological rigor to explicitly acknowledge the possibility of recall bias as a limitation. Additionally, it may be helpful for the authors to clarify how they addressed this in data collection or analysis (e.g., triangulation, probing recent events, or reflective journaling by the interviewer). (Page 13 for instance. Please consider adding line numbers in future manuscript versions to support more precise referencing and constructive feedback during peer review).

Discussion:

- The discussion offers important interpretations grounded in lived experiences of people with disabilities. However, I recommend the authors reflect more critically on the limitations of their study to enhance transparency and contextual integrity. While I understand that design elements cannot be altered at this juncture, these limitations can be acknowledged and leveraged to inform the interpretation of findings and shape future research.

- These may include:

- Scope Ambiguity: It remains unclear whether participants were referring to general healthcare access (e.g., for non-disability conditions) or specifically to disability-related services (e.g.,

rehabilitation). Clarifying this distinction—or at least acknowledging the ambiguity—would help readers better assess the applicability of the findings and recommendations.

- Recall Bias: Several participants shared experiences from early life, sometimes decades ago. This introduces the risk of recall bias, particularly for emotionally charged or stigmatizing events.

While this is a common challenge in qualitative research, acknowledging it would enhance the methodological transparency.

- Lack of Triangulation with Provider or Policy Perspectives: Including insights from healthcare workers or administrators could have enriched the analysis, especially regarding health system constraints

and policy feasibility. I suggest acknowledging this as a limitation and a direction for future work.

- Framework Fit: While the Missing Billion Framework was well-applied, the manuscript could more critically discuss its limitations—particularly around overlapping categories (e.g., affordability vs.

financing) and potential blind spots regarding intersectional factors (gender, poverty, urban ways of life). Addressing this would show a deeper engagement with the chosen analytical lens.

Conclusion:

- Strong and policy-relevant. However, a final sentence that ties the findings back to specific SDG goals may help reinforce global significance.

Reviewers' comments:

Reviewer's Responses to Questions

**Comments to the Author**

1. Does this manuscript meet PLOS Global Public Health’s publication criteria ? Is the manuscript technically sound, and do the data support the conclusions? The manuscript must describe methodologically and ethically rigorous research with conclusions that are appropriately drawn based on the data presented.

Reviewer #1: Yes

2. Has the statistical analysis been performed appropriately and rigorously?

Reviewer #1: N/A

3. Have the authors made all data underlying the findings in their manuscript fully available (please refer to the Data Availability Statement at the start of the manuscript PDF file)?

Reviewer #1: Yes

4. Is the manuscript presented in an intelligible fashion and written in standard English?

Reviewer #1: Yes

5. Review Comments to the Author

Reviewer #1: Thank you for the opportunity to review this important and timely manuscript. I have no addition comment. The manuscript is written in a standard English, and I have compiled all my comments in the uploaded document.

6. PLOS authors have the option to publish the peer review history of their article (what does this mean? ). If published, this will include your full peer review and any attached files.

**Do you want your identity to be public for this peer review?** For information about this choice, including consent withdrawal, please see our Privacy Policy .

Reviewer #1: No

---

## [Editor Report · Decision Letter 1]

12 Sep 2025

Left behind in primary healthcare: A qualitative exploration of healthcare experiences of people with disabilities in Ethiopia

PGPH-D-25-01404R1

Dear Mr Debalkie,

We are pleased to inform you that your manuscript 'Left behind in primary healthcare: A qualitative exploration of healthcare experiences of people with disabilities in Ethiopia' has been provisionally accepted for publication in PLOS Global Public Health.

Best regards,

Damen Haile Mariam, MD, MPH, PhD

Academic Editor